# Assessing Virtual Reality Spaces for Elders Using Image-Based Sentiment Analysis and Stress Level Detection

**DOI:** 10.3390/s23084130

**Published:** 2023-04-20

**Authors:** Makrina Viola Kosti, Nefeli Georgakopoulou, Sotiris Diplaris, Theodora Pistola, Konstantinos Chatzistavros, Vasileios-Rafail Xefteris, Athina Tsanousa, Stefanos Vrochidis, Ioannis Kompatsiaris

**Affiliations:** Centre for Research and Technology Hellas, CERTH-ITI, 57001 Thermi, Thessaloniki, Greece; nefeli.valeria@iti.gr (N.G.); diplaris@iti.gr (S.D.); tpistola@iti.gr (T.P.);

**Keywords:** visual analysis, sentiment, stress, virtual reality, elders, fusion

## Abstract

Seniors, in order to be able to fight loneliness, need to communicate with other people and be engaged in activities to keep their minds active to increase their social capital. There is an intensified interest in the development of social virtual reality environments, either by commerce or by academia, to address the problem of social isolation of older people. Due to the vulnerability of the social group involved in this field of research, the need for the application of evaluation methods regarding the proposed VR environments becomes even more important. The range of techniques that can be exploited in this field is constantly expanding, with visual sentiment analysis being a characteristic example. In this study, we introduce the use of image-based sentiment analysis and behavioural analysis as a technique to assess a social VR space for elders and present some promising preliminary results.

## 1. Introduction

The technological advancements of the last century have brought societies to face new realities that have incontrovertibly improved everyday life, making it more convenient and interesting. In recent decades, technology has made a true quantum leap with virtual reality (VR) and wearable devices that have been introduced and have already affected every area of life, with noticeable impact in the domains of education, tourism, healthcare, sports, entertainment, architecture and construction, etc.

VR is an emerging technology becoming more affordable, powerful, and robust every passing day, which can provide a sense of presence in the virtual world. This can be accomplished through visual and auditory cues that respond to the user’s actions, overcoming time and space limitations [1]. Thanks to these attributes, modern virtual environments, initially used only for gaming purposes, are now being exploited for a variety of tasks, such as sports activities, simulation, or even surgical techniques [2]. Moreover, VR is being used for rehabilitation, modernising the clinical approach for people with various diseases and conditions, such as for those undergoing rehabilitation after a stroke [3,4,5] or with cerebral palsy [6], or for other reasons such as physiotherapy [7] or orientation [8]. These health applications of VR have gained attention in recent years because conventional exercise methods in rehabilitation facilities are time- and space-consuming and expensive, as they need effort from human health professionals [9]. For example, VR applications and headsets are being used in care homes for older people suffering from Alzheimer’s, in order to improve their general mental health and trigger back memories [10,11]. Others examine the effect of immersive virtual reality (IVR) systems to improve balance and reduce fall risk in the elderly population [12].

VR applications appeal to a variety of age groups, such as adolescents, adults, and older adults (seniors). As the percentage of the elderly is increasing due to the rise in life expectancy, over the last few years, there has been an intensified interest in the implementation of VR tools addressed to the elderly for multiple purposes. One of them regards social isolation, which leads to the perceived feeling of loneliness. Social isolation has serious, so far underappreciated, public health risks that affect a significant percentage of older adults, according to the 2020 report of the National Academies of Sciences, Engineering, and Medicine (NASEM) [13,14]. The report stated that the elderly are at increased risk for social isolation because they tend to face factors such as living alone, loss of family or friends, chronic illnesses and other disabilities, or in some cases, mandatory isolation due to unforeseen external factors, such as the COVID-19 pandemic. It is a fact that during COVID-19, the extent of social loneliness was exposed among older adults. A plethora of studies investigate the psychological impact of social isolation, stating that the anxiety levels of the elderly increase due to the lack of social interaction. As a result, they become lonely, bored, exhausted, and distressed, which increases their depression levels and the risk of dementia [15,16,17,18,19].

At present, and due to accessibility and the improvement in technology, VR provides a more reachable option for older people to become engaged in social activities from the comfort of their homes as they are immersed in the three-dimensional simulation [20]. Although there are various commercial VR products on the market aimed at the elderly, such as Alcove (https://www.oculus.com/experiences/go/2855012877858033/?locale=en_US, accessed on 31 March 2023), for the social experience, or Appility (https://www.oculus.com/experiences/go/1465930903478165/?locale=en_US, accessed on 31 March 2023), Wander (https://www.oculus.com/experiences/go/1887977017892765/?locale=en_US, accessed on 31 March 2023), National Geographic (https://www.oculus.com/experiences/go/2252817104759749/?locale=en_US, accessed on 31 March 2023) for travelling, or Rendever (https://www.rendever.com/, accessed on 31 March 2023), addressing the issue of social isolation, they are mostly designed with limited formal consideration of the expectations of the elderly. Moreover, the evaluation of these tools has mostly been through surveys, usability questionnaires [21], or approaches based on verbal comments that rely upon user responses, Twitter data [22], etc. Finally, a few studies use EEG in the VR development process [23], which is difficult to apply in cases where the VR tool concerns a vulnerable social group, such as the elderly.

The goal of this work is to extend the current evaluation quiver for VR environments in academic research, using visual sentiment analysis and stress level detection as a means to assess a social metaverse for elders called “Cap de Ballon”. The social metaverse for elders assessed in this research paper was created in the context of a Horizon 2020 project called MindSpaces (https://mindspaces.eu/pilot-use-cases/, accessed on 31 March 2023). The proposed assessment methodology was implemented in the last stage of the prototype development of “Cap de Ballon”, which regarded a co-creation procedure between end users, artists, software engineers, and architects. Despite the small size of the sample used in our experimentation and the lack of statistical inference, we believe that our work lays the groundwork for further investigation and enrichment of the assessment of virtual environments using image-based sentiment analysis [24].

In the next section (Section 2), we provide a brief literature review regarding (a) digital environments developed for older adults and their evaluation approaches, and (b) image sentiment analysis approaches. Section 3 analyses the materials and methods exploited in our research work, followed by Section 4 with some preliminary results. Finally, Section 5 presents some conclusions and future work for our study.

## 2. Literature Review

### 2.1. VR Environments for Elders

There is a variety of developed VR experiences in the literature designed for the elderly. In ref. [25] the authors aim to assess the effectiveness of 360° immersive virtual reality interventions through videos on the well-being of older adults with or without cognitive impairment, as well as their attitudes towards this technology. The review analysed 10 articles published before April 2022 and found that VR 360° video interventions seem feasible, safe, and enjoyable for older adults in community-dwelling or residential aged care facilities.

Regarding the evaluation of VR environments, Rose et al. [26] tried an HMD-VR with people with dementia, which allowed patients and caregivers to navigate in pre-selected virtual environments. The evaluation of the VR was conducted via interviews and reports, revealing that the users were excited with the application, and the experience as a whole was beneficial since patients with dementia experienced more pleasure during and after HMD-VR compared to before exposure. In Matsangidou et al. [27], the authors propose an experimental design to investigate the feasibility of using VR technologies for the rehabilitation of patients with moderate to severe dementia. The authors, at the end of the paper, report the challenges faced during the experimental design, development, and implementation. Moreover, in ref. [21], the authors present a VR-based approach to allow elderly users to overcome problems related to social isolation. The evaluation in this study was conducted using interviews and questionnaires as well.

Additionally, concerning the use of VR in older populations, the authors of [28] investigated the acceptance and use of VR technology by older adults as a tool for active ageing. Thirty older adults were asked to use one of nine chosen VR applications twice a week for six weeks, and then fill out a questionnaire evaluating their acceptance of VR technology. The results showed that perceived usefulness, perceived ease of use, social norms, and perceived enjoyment had significant effects on the intention to use VR, and that VR was considered to have high acceptance among the elderly population. The study concludes that older adults have positive perceptions towards accepting and using VR for active ageing, as they perceive VR to be useful, easy to use, and an enjoyable experience. Ref. [29] also explores the use of VR in engaging older adults in Residential Aged Care Facilities (RACF). The study involved five RACF residents and five RACF staff members, who evaluated a VR system for two weeks. In this work, the researchers also utilise multiple interviews, research notes, and video recordings made during the VR sessions in order to evaluate the use of VR. The study found that the usability of interactive VR technology was affected by the ability of the aged care residents, especially those living with dementia. Additionally, the study identified that VR technology can engage older residents who may otherwise isolate themselves. Overall, the study highlights the potential benefits of using VR technology in aged care, as well as the need for design improvements to ensure its effective use with older adults. Other studies have focused on spatial orientation, such as [8], in which the authors investigate whether the decline in spatial orientation with age is similar in virtual reality (VR) as it is in the real world; such studies are important, as they could influence the efficacy of VR tools for older people, particularly in physical therapy. The study showed a similar influence of VR on the spatial orientation of both age groups, supporting the usage of VR regardless of age. Moreover, the authors in [30] aim to evaluate a novel social VR platform that pairs older adults from different locations to engage in virtual travel and productive activities together, collecting data through interviews. The results showed that the VR was feasible and acceptable for older adults, including those with cognitive impairment. Perceived spatial presence was found to be a significant factor in the program’s success, and participants reported high levels of engagement and enjoyment. The study suggests that VR social applications may foster social engagement among older adults.

Rendever [31], on the other hand, is a commercial VR application that gives elderly people the possibility of immersing in and navigating virtual worlds using a range of customised VR hardware and content in the form of 360-degree videos. The acceptance of Rendever, on the one hand, is proof of the potential of using VR by the community of elderly people, and on the other hand, highlights the need to study and develop methodologies for evaluating such environments so that they best meet the needs of the elderly.

In this article, we utilise a VR social platform developed in the context of a European project in which the authors participated, which exploits the potential of combining art, technology, AI, and VR to preserve and improve neurological, cognitive, and emotional functions while considering mental health. In relation to the proposed literature on social platforms for elders, the use of a social platform with these characteristics per se is a novelty. For instance, artists can use interactive installations and design environments that utilise VR, visual analysis, and AI algorithms to create spaces that elicit positive cognitive and emotional responses and memories. By tailoring these environments to the preferences and behaviours of the end-users, artists can stimulate seniors intellectually and evoke positive emotions and memories, which in turn can encourage communication and address the emotional and cognitive needs of seniors.

The scope of this paper is to take the evaluation process of such environments one step further. Therefore, we introduce a novel approach in the field, which aims to assess VR spaces for elders by exploiting the prospects of image sentiment analysis and behavioural analysis.

### 2.2. Image Sentiment Analysis Approaches

Image sentiment analysis can facilitate the designing of multimodal systems able to evaluate human emotional stimuli exposed to three-dimensional spaces. However, all sentiment extraction tasks foster risks because of the subjectivity that governs human emotions [32]. 

Utilising Convolutional Neural Networks (CNNs) [33,34,35,36] is an effective way to extract a viewer’s emotional stimulus. Different approaches to sentiment expression have been presented in the literature [32], which fall into two main categories for emotion modelling, namely the Dimensional Model and the Categorical Model. The Dimensional approach represents emotions as points in a 2D or 3D space. Emotions have three basic dimensions, namely valence, arousal, and control.

Due to the small effect of the control dimension, also known as dominance, most of the related literature focuses only on valence and arousal (see Figure 1).

Some studies only consider the valence dimension, predicting the sentiment polarity in terms of two levels (positive and negative), three levels (positive, neutral, and negative), or even more, like in [38], which uses a five-level method. On the other hand, the Categorical Model can be used to map particular emotions such as “happiness”, “anger”, etc., in the valence–arousal–control space [39,40].

A variety of image sentiment analysis methods have been suggested in the literature. Early works use low-level image features (such as colour or texture), whereas more recently, semantic features, machine learning, and deep learning methods are most commonly used. The idea of Adjective–Noun Pairs (ANPs) was first introduced in [41] to describe the analysed images in terms of emotions or sentiments. In the same work, a bank of visual classifiers, known as SentiBank, was suggested to identify 1200 ANPs that are present in pictures. Sentribute was introduced in [42] as an alternative to low-level attributes for categorising visual emotions. The authors were successful in connecting these characteristics to the emotions elicited by the images.

Support vector machine (SVM) classifiers were exploited by the authors of [43] to extract image features from ANPs for sentiment classification, achieving a precision of 0.86 on the Visual Sentiment Ontology (VSO) dataset. ANPs used for automatic emotion or sentiment recognition produce better results than low-level image descriptors [44]. The authors of [45] suggest the usage of local information instead of improving holistic representations based on the observation that both the entire image and local regions contain important sentiment information. In [46], the researchers introduce the deep-coupled adjective and noun network (DCAN), a novel CNN architecture for image sentiment analysis, which yields better results than those of earlier studies that had used ANPs. Using various image datasets, with or without combining the activation maps of the convolution layers with SUN [47] and YOLO [48] semantic attributes, five different CNN architectures—four commonly used in machine learning and one specifically created for sentiment analysis—were compared in [49]. In the same study, a brand-new urban outdoor image dataset called OutdoorSent was also suggested.

In this work, we adopt the Dimensional Model with both the valence and arousal dimensions and a three-level sentiment polarity, viz, three classes for valence (positive, neutral, and negative) and arousal (excited, neural, and calm), as shown in Figure 1.

## 3. Materials and Methods

In our study, we use image-based sentiment analysis and behavioural analysis to assess “Cap de Ballon”, a social metaverse developed with the elders for the elders during MindSpaces, a Horizon 2020 project. In the context of social VR assessment, the tools mostly used are user reports, interviews, or sentiment analysis based on text. Recently, the advances in deep learning, as well as the distribution of multimedia through social networks, have increased the focus on image sentiment analysis research. To evaluate VR environments, image sentiment analysis can be utilised to analyse the emotional responses of users when immersed in virtual spaces. One possible method involves using cameras or sensors to track the facial expressions and body language of users as they interact with the VR environment. The collected data can then be processed using machine learning algorithms to determine the user’s emotional state.

Another approach involves the use of natural language processing (NLP) techniques to analyse users’ feedback and comments after a VR experience. This involves requesting users to complete surveys or provide feedback on their emotional state while using the VR environment. The obtained feedback can then be analysed using sentiment analysis algorithms to assess the overall emotional response of users.

The utilisation of image sentiment analysis for evaluating images of a VR environment and the sentiment evoked by them has not been used in the context of VR evaluation. It can provide valuable insights into how users are experiencing the environment and help developers identify areas for improvement. For instance, if the sentiment analysis indicates that users feel anxious or uncomfortable in certain parts of the VR environment, developers can utilise this information to enhance the user experience.

Because of the nature of the social group we are addressing, there was difficulty in involving a large number of elders in our experiment and in repeating the data collection process. Therefore, our experiment regards only one experimentation session, in which we gathered our subjects to immerse them in our Virtual Village, while we collected data on the most visited places during navigation in terms of time spent in a given point of the virtual space. The rationale behind our experiment was to collect navigation information regarding the most visited places and analyse them by exploiting visual sentiment analysis and behavioural analysis. Briefly, the idea is presented in Figure 2.

### 3.1. Subjects

Our target group was comprised of ten (10) autonomous seniors aged between 60 and 85 years living independently in Paris, France, who did not have particular health issues or mental illnesses. All participants were considered non-vulnerable.

Informed consent was obtained from all subjects involved in this study. Before conducting the evaluation/experimentation session, we ensured all participants signed informed consent forms, which materialised the decision to participate. The documents provided were clear, straightforward, and ensured that participants understood and agreed with data processing and procedures before participating.

Our research team provided all necessary clarification and answered all questions the subjects had before participating in the research study. Precise information on data collected, purpose, data processing, and storage was available in the consent information sheets provided before every experimentation cycle.

### 3.2. Image Sentiment Analysis

Image sentiment analysis involves the prediction of the emotional response that an image elicits from its viewers. As social networks and VR applications grow in popularity, image sentiment analysis has become an important area of study for both academia and industry. The ability to automatically understand the emotions conveyed by images and videos has many potentials and applications, including online advertising, brand monitoring, and customer feedback. Our methodology focuses on analysing the emotions that both real and synthetic images of indoor and outdoor spaces evoke in viewers. In this study, we used screenshots of a social VR space for elders. To analyse these emotions, we employed the dimensional sentiment model, which considers both the valence and arousal dimensions. We then used a three-level sentiment polarity system to classify the emotional response, with “positive”, “neutral”, and “negative” for valence and “excited”, “neutral”, and “calm” for arousal.

#### 3.2.1. Our Dataset

To train and evaluate the produced visual sentiment analysis CNNs, 1064 images were selected from different sources, including the Places dataset [50], screenshots from VR environments, and images obtained through crawling. Each one of these images was annotated by at least five individuals in terms of valence and arousal, following a similar approach to [51]. A total of 50 people participated in the annotation task, with the annotators’ group comprising individuals with backgrounds in architecture and design, as well as European citizens of varying ages. It is important to note that, to our knowledge, this is the first dataset for sentiment analysis of images depicting indoor and outdoor spaces that include both real and synthetic images.

#### 3.2.2. The Method Used

During our research, we experimented with Convolutional Neural Network (CNN) architectures that are widely used for image analysis tasks, namely VGG-16 [33], InceptionV3 [52], ResNet50 [34], DenseNet169 [35], and Xception [36], adapting them to address the needs of our problem by training them on the annotated dataset that we created. In addition, we examined two architectures with low complexity compared to the above, the Robust architecture [51], which was previously tested on sentiment analysis of images, and advanced Vision Transformers (ViT) [53] that have not been used in the field of image sentiment analysis.

The final algorithm for image sentiment analysis was developed based on two fine-tuned VGG-16 architectures, one used for predicting valence and the other for predicting arousal. A simplified image sentiment analysis pipeline is presented in Figure 3. The deployed algorithm has shown a predictive ability of approximately 61% and 60% accuracy for valence and arousal, respectively, evaluated using our dataset [24]. This is a commendable performance given the subjectivity of the problem and considering how the dataset we used is quite demanding, since it contains both real and synthetic images of indoor and outdoor spaces. In [49], VGG-16 achieves about 65% mean accuracy for valence using two classes (positive and negative), while the dataset used contains only real outdoor images. There are no similar works on image sentiment analysis for arousal, so we cannot have any comparison concerning the performance of the arousal network.

Throughout our experiments, we trained and evaluated the aforementioned network architectures using our dataset. The best-performing network for both valence and arousal was DenseNet169 [35], achieving an accuracy of around 68% and 70%, respectively [24]. Although the VGG-16-based architectures did not have the best performance in the quantitative evaluations, the final selection was made by also taking into account some qualitative evaluations, where the algorithm produced more coherent results and had better generalisation ability. More specifically, the deep learning models were also assessed based on their ability to predict similar sentiment results for similar images, as it was assumed that comparable visual characteristics should evoke the same sentimental response. For this qualitative assessment, we used a variety of real images and VR screenshots of the same content (e.g., images of the same buildings or urban areas), and we compared the valence and arousal results for similar views (see Figure 4 and Figure 5). The images presented in Figure 4 and Figure 5 were captured in the context of the MindSpaces Horizon 2020 project for the reconstruction of the corresponding 3D model of the depicted building to insert it in VR environments.

Finally, the algorithm and the overall image sentiment analysis methodology were tested and validated in other use cases as well, such as in urban environments [54] and indoor space design [24].

### 3.3. Behavioural Analysis

For this study, a behavioural data analysis module was also implemented, which was developed and validated in [55]. The behavioural data analysis consists of a feature extraction process and a Hidden Markov Model (HMM) to acquire stress levels from the positional data of the users (see Figure 6). We extract four different features from the positional data, namely moving time, track spread, wandering style, and hotspot. Moving time is the total time that the user is moving above a certain speed threshold. Track spread is the total distance of every point of the user’s trajectory from the centre of the trajectory. For the next two features, we define square cells of the movement of 0.1 metres to the side. The wandering style is the number of unique cells in the trajectory divided by the total distance covered. Finally, the hotspot feature is the distance of the most visited cell from the centre of the trajectory. Using these features and feeding them to an HMM, we are able to compute stress levels based on the user’s positional data.

More specifically, the method followed is separated into the feature extraction and stress prediction steps, based on the work of [55], but with some modifications to fit our scenario. As we already mentioned, the features extracted include the moving time, track spread, wandering style, and hotspot spread. Moving time (Algorithm 1) is the total time the subject was moving with a speed above a certain threshold.
(1)mt=∑tv>0.2 ms,

This threshold was set to 0.2 m/s. This feature is useful for stress detection since it can reveal the time the subject was in a rush, which typically occurs when people are nervous.
**Algorithm 1** Computation of moving time
**Data:** User’s trajectory data (pos = (x, y), array), timestamps (t, array)
**Results:** Moving time (mv) feature**1**move_time_array ← [], i = 1**2****while** i < length(pos) **do****3** v ← √(pos_i_(x) − pos_i-1_(x))^2^ + (pos_i_(y) − pos_i-1_(y))^2^/(t_i_ − t_i−1_)**4**
**if** v > 0.2 **then:****5**
  move_time_array ← (t_i_ − t_i−1_)**6**
**end****7****end****8**mv ← mean(move_time_array)**9****end**

*Track spread* (Algorithm 2) refers to the maximum distance between any point of the trajectory of the subject and the centre of its trajectory.
(2)$$ts=\underset{i=\left(0,t\right)}{\max}\left(dist\left(Tr_{c},Tr_{i}\right)\right),$$
where $Tr_{i}$ is the point of trajectory in the timestamp i and $Tr_{c}$ is the centre of the trajectory. This feature provides information regarding the quantity of space the subject occupies.
**Algorithm 2** Computation of track spread
**Data:** User’s trajectory data (pos = (x, y), array)
**Results:** Track spread (ts) feature**1**track_spread_array ← [], i = 1**2**track_center ← (mean (pos(x)), mean (pos(y)))**3****while** i < length(pos) **do****4**  distance ← √(pos_i_(x) − track_center(x))^2^ + (pos_i_(y) − track_center(y))^2^**5**
 track_spread_array ← (t_i_ − t_i−1_)**6****end****7**ts ← max (track_spread_array)**8****end**

For the other two features, the computation of cells is first needed, which are of squared shape with a size of 0.1 m. *Wandering style* (Algorithm 3) is the fraction of the number of unique cells divided by the total length of the subject’s trajectory (Equation (3)).
(3)$$ws=\frac{cells_{number}}{traj_{length}},$$

This feature also reveals how much space the subject occupies but without taking care of the small movements within a cell. It also shows the way the subject moves within the space, for example, making circles around certain areas or having a more straight-line trajectory.
**Algorithm 3** Computation of wandering style
**Data:** User’s trajectory data (pos = (x, y), array)
**Results:** Wandering style (ws) feature**1**cells ← unique((round (pos(x)), round(pos(y))))**2**i = 1, trajectory_length = 0**3****while** i < length(pos) **do****4**  distance ← √(pos_i_ x) − pos_i−1_(x))^2^ + (pos_i_(y) − pos_i−1_(y))^2^**5**
 trajectory_length ← trajectory_length + distance**6****end****7****if** trajectory_length > 0 **then****8**
 ws ← length (cells)/trajectory_length**9****else****10**
 ws ← 0**11****end****12****end**

The last feature, *hotspot spread* (Algorithm 4), is the distance between the centre of the most visited cell, also called the hotspot of the trajectory, and the centre of the trajectory.
(4)$$hs=dist\left(H_{c},\;Tr_{c}\right),$$
where $H_{c}$ is the centre of the hotspot of the trajectory and $\;Tr_{c}$ is the centre of the trajectory. This feature provides knowledge regarding the most visited cells in the trajectory.
**Algorithm 4** Computation of hotspot spread
**Data:** User’s trajectory data (pos = (x, y), array)
**Results:** Hotspot spread (hs) feature**1**cells ← unique((round (pos(x)), round (pos(y))))**2**hs_center ← (mean (cells(x)), mean (cells(y)))**3**track_center ← (mean (pos(x)), mean (pos(y)))**4**hs ← √(track_center(x) − hs_center(x))^2^ + (track_center(y) − hs_center(y))^2^**5****end**

The feature extraction was performed in a time window of 5 s. After features are extracted, they are fed into an HMM for the computation of arousal level or stress, which varies between six discrete levels, from very low to very high.

### 3.4. Fusion of Sentiment Arousal and Behavioural Stress

After acquiring the most visited cells, screenshots were taken from the cell in every direction in the virtual space, and the screenshots were processed from the image-based sentiment analysis method presented in Section 3.2. For the fusion of the stress or arousal score from the behavioural data analysis with the results from the image-based sentiment analysis, the discrete levels were categorised into numbered values in the range of 0–1. The final fusion was a modified probability averaging between the behavioural data analysis results and the visual data analysis results, where the visual analysis results were first averaged alone and then fused with the behavioural data results.

### 3.5. Cap de Ballon

The Collaborative VR Application was designed for multiple users who could immerse into a single VR environment. The VR Application was built using Steam VR to provide maximum compatibility with VR devices. The application allowed users to log into their user accounts and create meeting rooms, which other users could join. A “VR Ready” laptop was also able to run the tool. Moreover, the VR meeting room server could host a multiplayer server that allowed multiple users to interact with each other. The server also populated the VR environment with the digital artefacts provided by a backend database. The VR meeting room server was hosted on a dedicated server that allowed us to build and run the service.

In general, Cap de Ballon (Figure 7 and Figure 8) is a 3D VR village, the product of a co-creation design process having as main goal to stimulate seniors’ memory and improve social connection in order to fight social isolation. It is built upon a large-scale rocky context, surrounded by water, and has whitewashed houses that reference Greek Cycladic architecture. The village is organised into four thematic categories, which were co-decided with the subjects, and each neighbourhood can be distinguished by colour. Seniors contribute to the co-creation aspect of the village with content produced by them, such as videos, pictures, and slideshows, which are displayed on the windows of the houses of this social metaverse. The platform provided the possibility for seniors to communicate with others inside VR, teleport to other windows and neighbourhoods, express their emotions through emoticons, and interact with a variety of objects.

## 4. Results

To evaluate the sentiment evoked by the VR environment during the experimentation phase, a procedure was implemented as described in the previous sections (see Figure 9). Arousal (or intensity) is the measurement that represents the level of *activation* an experience induces, ranging from calm (or low) to normal (or medium) to excited (or high), while valence measures the degree of pleasantness, ranging from negative to positive. In our experiments, results in the range of [0,0.33) are equivalent to calm/negative arousal/valence, values in the range of [0.33,0.66) are equivalent to neutral arousal/valence measurement, and values in the interval [0.66,1] correspond to excited/positive arousal and valence, correspondingly.

The results of each one of the sentiment indexes (fused arousal and valence) for the 10 subjects that participated in our experiment are shown in Table 1 and Table 2, and the graphical representation of these results is depicted in Figure 10a,b. From the results, it can be seen that most of the subjects had a relatively positive experience, as indicated by the valence scores. More specifically, out of the 10 subjects, 2 subjects had low valence scores, that is, below 0.3; 2 subjects had neutral valence scores, that is, scores between 0.3 and 0.6; the remaining 6 subjects had positive valence scores, that is, scores over 0.6. The results also indicate that the experience of all subjects can be regarded as calm, since all the subjects had arousal scores below 0.5.

Additionally, Figure 11 represents the mean values of valence and fused arousal per subject over all five (5) most visited hotspots. From the representation of valence and arousal in Figure 11, we are able to observe that our subjects, in reference to the most visited places (hotspots), were in general activated (subjects 1, 3–5, and 7–10), feeling calm during the exploration of the village, and pleasant (subjects 2–3 and 5–9).

A more detailed representation is shown in Figure 12a–e, which describe the valence and fused arousal scores for a single subject in each one of the five points he/she visited more during the experiment.

## 5. Discussion and Conclusions

In this paper, we investigated the possibility of exploiting the prospects of using image-based sentiment analysis and behavioural analysis as main or supplementary tools for quantifying the emotion of a user in relation to the acceptance of a virtual space.

Despite the satisfactory group size of participants for a preliminary study, we faced difficulties in accessing more subjects due to the age group we were targeting. Therefore, this study consequently included only 10 participants who did not have any health issues or mental illnesses, lived independently, and were aged between 60 and 85 years old.

Although our subjects had diverse profiles, expectations, and varying attitudes towards social platforms, they all shared a willingness to try a new VR experience, as they perceived it as a way to broaden their perspectives and perceptions. As it was their first time using a VR social platform, they found it challenging to communicate with others while navigating and using the microphone simultaneously. This means that the overall VR experience was in some amount affected by the subjects’ level of mastery of the system and its commands. However, we managed to some extent to smooth out the effect of lack of experience in using virtual environments by providing users with a detailed explanation tutorial of the VR environment functionalities. The participants recognised the great potential of the VR metaverse but acknowledged the need for a tutorial to better understand and master the joysticks and other features.

We described in detail the methods and techniques we used to analyse images taken from a social metaverse, *Cap de Ballon*, to assess the human emotional state induced by visual and sound stimuli in a VR space. Cap de Ballon is an art-driven co-creation between the final users (elders), architects, artists, and software engineers. Our results show that the method is able to capture the sentiments generated from the navigation data of the elders in the social virtual space. More specifically, during the experiment, we analysed the sentiment indexes (fused arousal and valence) of 10 subjects who explored a social VR village. The results indicate that most of our subjects had a positive experience, as indicated by their valence scores, with two subjects having low valence scores, two subjects having neutral valence scores, and six subjects having positive valence scores. All subjects had arousal scores below 0.5, which indicates that during their experience, they felt generally calm. The mean values of valence and fused arousal per subject over the five most visited hotspots can be seen in Figure 11. The figure shows that the subjects were generally activated and felt calm during the exploration of the village and pleasant. Overall, the subjects felt activated, calm, and pleasant during the exploration of the village, which indicates, at some level, the acceptance of the VR tool and its content to the end users.

The current study contributes to the growing body of research on the use of virtual reality technology among the elderly. More specifically, our work focuses on image-based sentiment analysis and behavioural analysis and how they can be utilised in the context of assessing VR spaces for elders. By analysing images of virtual environments, we can gain insight into the emotional responses of elders and use this information to improve the design and functionality of VR spaces to better meet their needs. This approach can be applied in a supplementary way, together with other instruments, such as verbal reports or questionnaires, interviews, EEG measurements, or sentiment analysis based on text. Alternatively, it can be recruited as a main vehicle for the post-use assessment of a VR space.

Furthermore, image-based sentiment analysis creates new prospects and potentials, as with proper implementation, it can also be utilised for real-time evaluation, with the goal of adapting VR environments according to the user’s emotional state in real time.

This study includes some limitations that need to be addressed. First of all, the number of participants is small, which in combination with the lack of advanced statistical inference, creates shortcomings in the generalisability of our results. Secondly, this study is limited only to one age group, which means that it refers only to a specific social cluster. To extend the proposed study to other age groups in the future, researchers could recruit participants from different age ranges and assess their emotional responses to the VR experience they are offered. This would make feasible the comparison between age groups and provide insight into the differences in emotional responses of users of different ages. This gives the possibility to consider exploring how the design and functionality of VR spaces may need to be adapted for different age groups. For instance, teenagers may have different preferences for virtual environments compared to middle-aged or elderly individuals. By assessing the emotional responses of participants from different age groups, researchers will be able to identify design features that are most effective at eliciting positive emotions and adapt the VR environment accordingly. In addition, a comparative analysis of VR-based and desktop-based experiments could offer valuable insights regarding the impact of the VR experience on the sentiments evoked in the users. Even so, the use of VR stimuli is known to increase the emotional response of the users while maintaining the controlled environment of a virtual world [56].

Even though our work is based on a small number of participants and lacks advanced statistical inference, we believe that it lays the foundation for further research. For instance, in order to be able to statistically compare the sentiment measurements with future ones, aiming to improve Cap de Ballon, our intention is to incorporate image sentiment analysis in the future development stages of this social metaverse. Additionally, we consider incorporating other measures of emotional response, such as self-reported questionnaires, or physiological measures, such as heart rate variability or skin conductance. This would provide a more comprehensive understanding of the emotional experience of participants and which can be extended for accurate comparisons between age groups. By integrating sentiment analysis measurements into a broader dataset, we will be able to perform a more detailed empirical study.

Finally, by further enriching our work by applying additional emotion extraction techniques, using physiological signal processing algorithms to detect users’ emotional states by measuring their physiological signals (i.e., GSR—Galvanic Skin Response), we will be given the possibility to optimise the adaptiveness of a VR environment in real time.

## Figures and Tables

**Figure 1 sensors-23-04130-f001:**
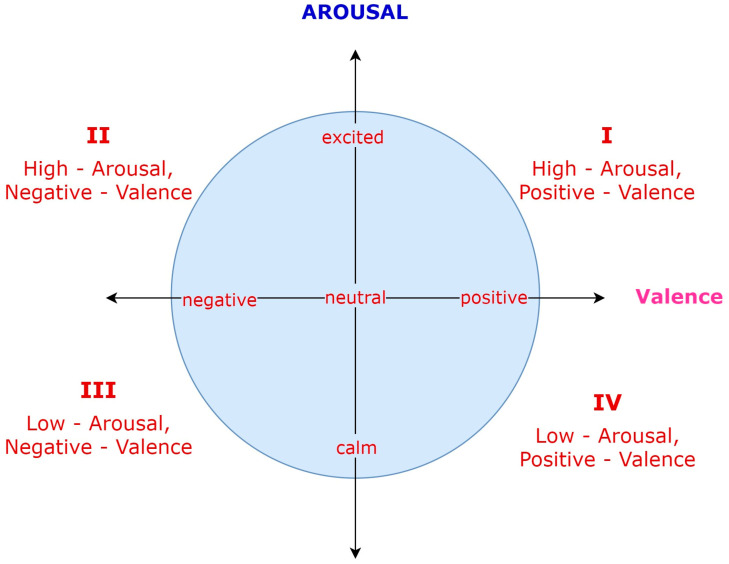
Two-dimensional valence–arousal representation, based on [37].

**Figure 2 sensors-23-04130-f002:**
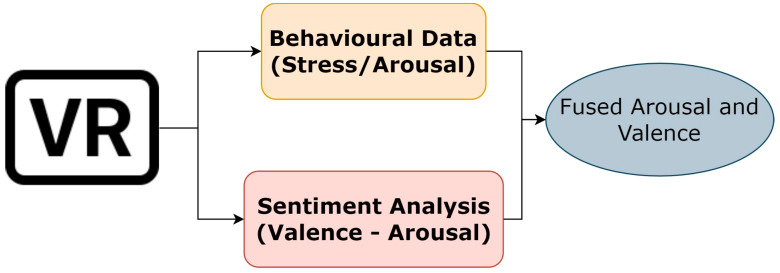
Image sentiment analysis and behavioural analysis fusion.

**Figure 3 sensors-23-04130-f003:**
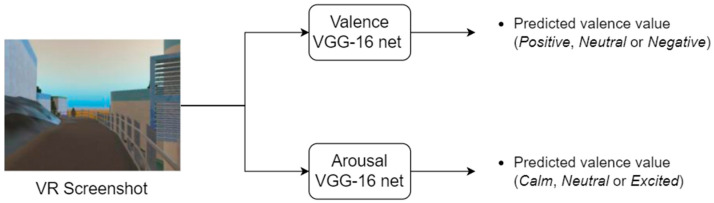
Simplified image sentiment analysis pipeline. The predicted valence and arousal values are fed to the fusion algorithm.

**Figure 4 sensors-23-04130-f004:**
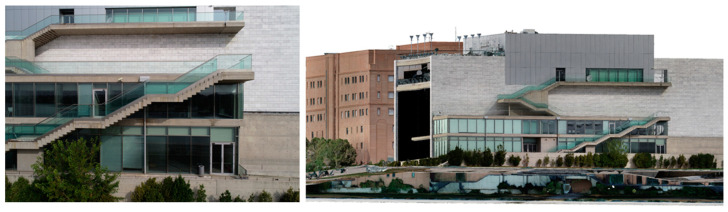
On the (**left**) we have a real image of a specific place, while on the (**right**) we have a screenshot from the corresponding 3D model that depicts a similar view as the real image with similar visual characteristics. The image sentiment analysis algorithm produces a “positive” result for valence for both images.

**Figure 5 sensors-23-04130-f005:**
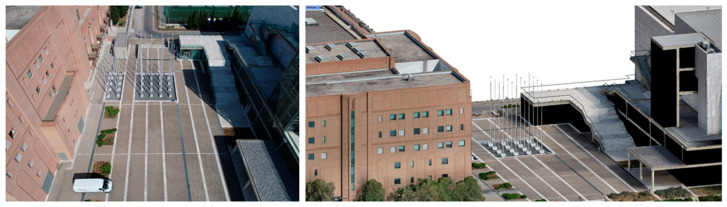
On the (**left**) we have a real image of a specific place, while on the (**right**) we have a screenshot from the corresponding 3D model that depicts a similar view as the real image with similar visual features. The image sentiment analysis algorithm produces a “neutral” result for valence for both images.

**Figure 6 sensors-23-04130-f006:**
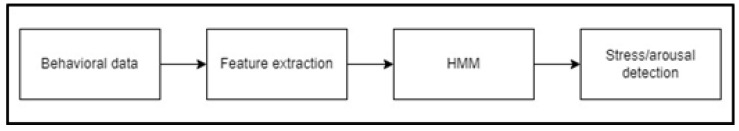
Workflow of the behavioural data analysis.

**Figure 7 sensors-23-04130-f007:**
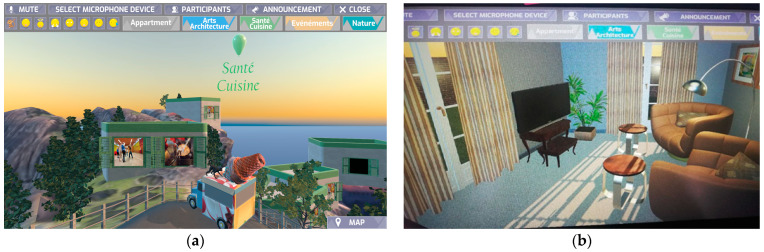
Cap de Ballon, (**a**) outdoors view and (**b**) indoors view.

**Figure 8 sensors-23-04130-f008:**
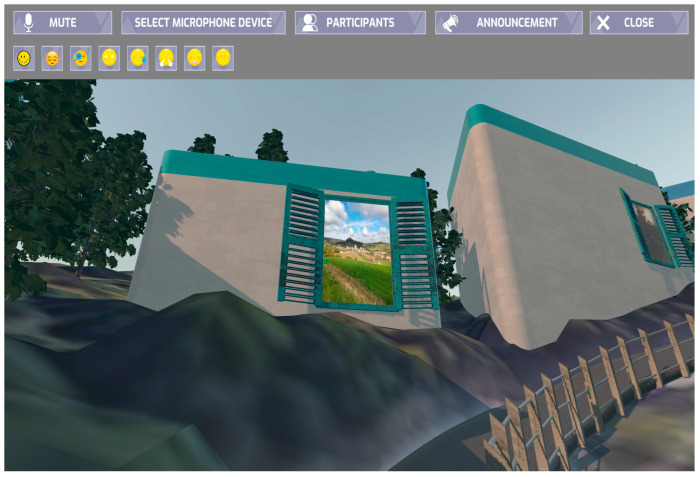
Cap de Ballon, content windows.

**Figure 9 sensors-23-04130-f009:**
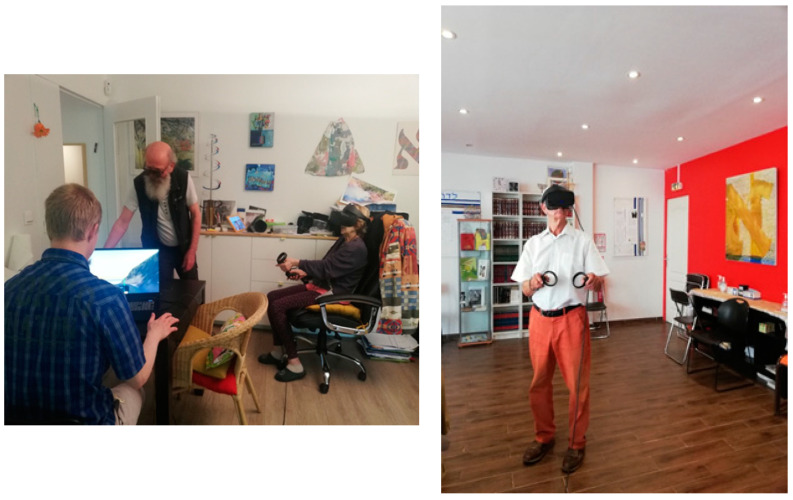
Photographic snapshots of the experimentation session with our elder subjects.

**Figure 10 sensors-23-04130-f010:**
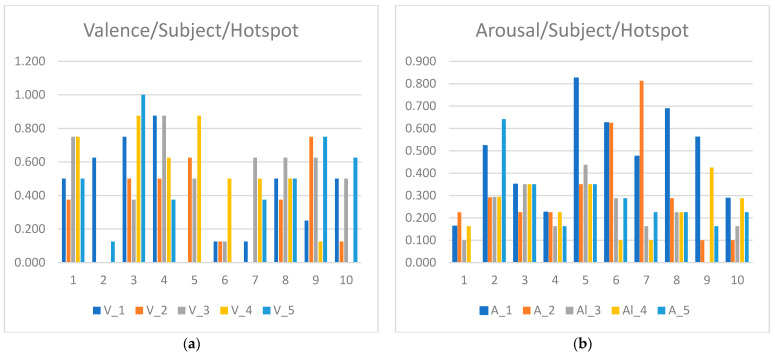
Graphical representation of the (**a**) valence and (**b**) fused arousal scores for each subject per hotspot.

**Figure 11 sensors-23-04130-f011:**
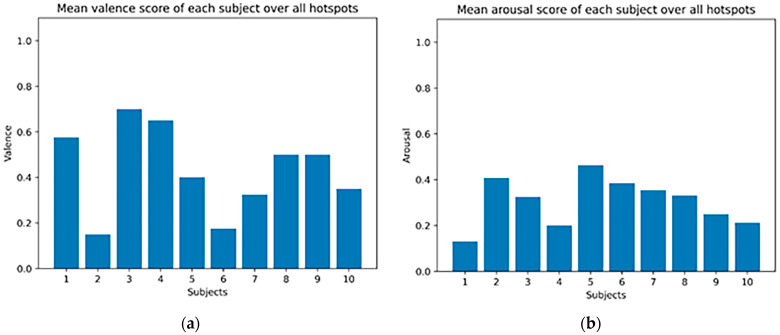
Mean (**a**) valence and (**b**) fused arousal per subject over all hotspots.

**Figure 12 sensors-23-04130-f012:**
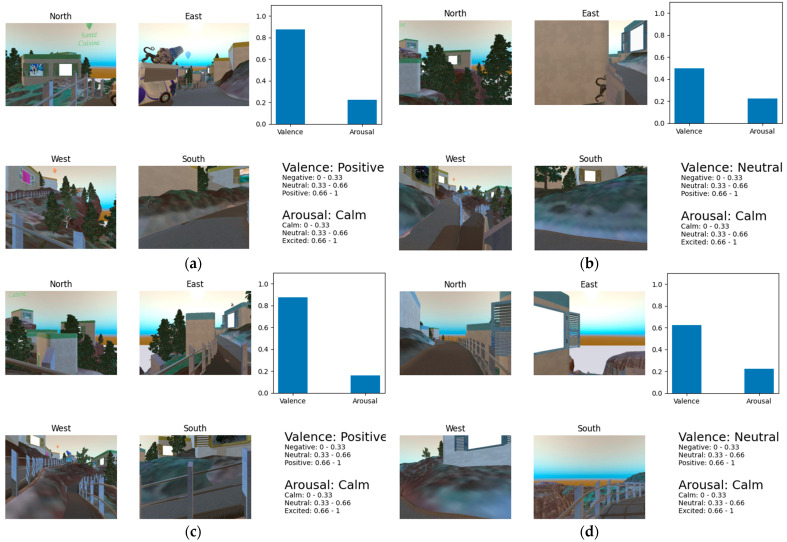
Valence and arousal scores for one subject per mostly visited spots (top 5). Each figure, from (**a**–**e**) represents the user’s sentiment per each one of the five hotspots.

**Table 1 sensors-23-04130-t001:** Valence scores per subject and per hotspot.

	V_1	V_2	V_3	V_4	V_5	Mean	SD
Subject_1	0.500	0.375	0.750	0.750	0.500	0.575	0.150
Subject_2	0.625	0.000	0.000	0.000	0.125	0.150	0.242
Subject_3	0.750	0.500	0.375	0.875	1.000	0.700	0.232
Subject_4	0.875	0.500	0.875	0.625	0.375	0.650	0.200
Subject_5	0.000	0.625	0.500	0.875	0.000	0.400	0.348
Subject_6	0.125	0.125	0.125	0.500	0.000	0.175	0.170
Subject_7	0.125	0.000	0.625	0.500	0.375	0.325	0.232
Subject_8	0.500	0.375	0.625	0.500	0.500	0.500	0.079
Subject_9	0.250	0.750	0.625	0.125	0.750	0.500	0.262
Subject_10	0.500	0.125	0.500	0.000	0.625	0.350	0.242
Mean Over Subjects	0.425	0.338	0.500	0.475	0.425		

**Table 2 sensors-23-04130-t002:** Arousal scores per subject and per hotspot.

	A_1	A_2	Al_3	Al_4	A_5	Mean	SD
Subject_1	0.163	0.225	0.100	0.163	0.000	0.130	0.076
Subject_2	0.523	0.291	0.293	0.294	0.641	0.408	0.147
Subject_3	0.350	0.225	0.350	0.350	0.350	0.325	0.050
Subject_4	0.225	0.225	0.162	0.225	0.163	0.200	0.031
Subject_5	0.825	0.350	0.438	0.350	0.350	0.463	0.184
Subject_6	0.625	0.625	0.288	0.100	0.288	0.385	0.208
Subject_7	0.475	0.813	0.163	0.100	0.225	0.355	0.262
Subject_8	0.688	0.288	0.225	0.225	0.225	0.330	0.180
Subject_9	0.561	0.100	0.000	0.425	0.163	0.250	0.210
Subject_10	0.288	0.100	0.163	0.288	0.225	0.213	0.073
Mean Over Subjects	0.472	0.324	0.218	0.252	0.263		

‘A’ refers to the fused arousal score derived from the image-based sentiment analysis and the behavioural analysis for stress detection.

## Data Availability

The developed and used dataset can be exploited by the research community, acting as a benchmark for image-based sentiment analysis and is publicly available following the required copyrights: https://m4d.iti.gr/urban-indoor-outdoor-sentiment-analysis-dataset-mindspaces/ (accessed on 31 March 2023).

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
