# Peer review of "Assessing Virtual Reality Spaces for Elders Using Image-Based Sentiment Analysis and Stress Level Detection"

_sensors, 2023, doi:10.3390/s23084130_

Round 1

Reviewer 1 Report

Authors discuss how VR can help elderly using imagery for sentiment and stress level detection. This article is nicely presented with a demand of such technology at current time. However few points must be improved as follows.

1. Literature review section needs improvement. Currently it lacks in-depth comparative analysis of existing research works.  It would be nice to include some technologies or company tools which do similar works.

2. Provide flowchart as well as algorithm for the methods sections. Currently it is not well to read.

3. Results sections needs improvement. Include the table or chart about the impact of VR for imagery on Stress level and sentiment analysis. Currently, the results section is not well designed. Put more insights into it.

4. In discussion section, you should include the results in more precise way. Also, mention how this study can be delved with other ages of people such as teenage, middle age etc.,

5. Improve references with more recent articles and technologies.

6. Also include consent approval form the participants. 

Reviewer 2 Report

This paper introduces the use of image-based sentiment analysis and behavioural analysis as a technique to assess a social VR space for elders and present some results.

The topic is relevant. There is new contribution from the paper.

The application is interesting. The conclusions are consistent with the evidence.

The paper lacks 2023 references. Please discuss more recent relevant work.

Please explain the choice of the evaluation criteria, such as, explain with technical details why selected criteria are important for this paper’s application. 

Explain in good technical details on complexity, validity and generalisability of the results.

Please explain limitation of the current solution.

In page 3, is Figure 1 own innovation?  If not, explain source please.

In page 5, is Figure 2 own innovation?  If not, explain source please.

In page 10, Table 1 and 2 adopted only 10 samples. Is the sample size big enough to have generalisable results?

Reviewer 3 Report

The authors introduce the use of image based sentiment analysis and behavioural analysis as a technique to assess a social VR space for elders. The main idea is interesting and very important in the current global context of increased life expectancy. The paper is very well writen with a few minor errors found and listed in sequence. Please note that the next comments are intended to improve paper quality and readers' understanding.

The sentiment and stress were both assessed using metrics pertaining to the experience of the elderly in the VR environment. I was wondering how previous experience they could have in virtual reality would influence the values measured. Please comment on that. It would be nice to have some information on the text whether any of the 10 users had previous experience with VR or how easy was to them controlling the virtual application.

"This algorithm has shown a predictive ability of approximately 61% and 60% accuracy for valence and arousal respectively" -> is this prediction rate good enough? It would be nice to have some comparison with state of the art techniques to show how close/far is the proposed technique to the best ones used.

"adapting them to address the needs of our problem" -> how were they adapted?

"For this qualitative assessment, we used a variety of real images and VR screenshots of the same content (e.g., images of same buildings or urban areas) and we compared the valence and arousal results for similar views (see Figures 3 and 4). " -> what is the source for those images and 3D models? Were they created by the authors themselves?

The authors show that are already plenty of VR solutions devoted to elderly people. Please make clear what are the differences of the proposed work to the already existing ones.

Important references are missing in the text, such as:

- Virtual reality among the elderly: a usefulness and acceptance study from Taiwan

- Evaluating the use of interactive virtual reality technology with older adults living in residential aged care

- Suitability test of virtual reality applications for older people considering the spatial orientation ability

- Using Immersive Virtual Reality to Enhance Social Interaction

among Older Adults: A Multi-site Study

More general comments and minor errors are listes as follows.

"paper, was created" -> "paper was created"

"by at least by five " -> "by at least five "

"CNN" -> please define the term only once in the text

"(ViT) [44] that has not used" -> "(ViT) [44], that has not been used"

"HMM" -> please define the term only once in the text

" as a main goal" -> " as main goal"

"users, artist, software" -> "users, artists, software"

Round 2

Reviewer 1 Report

This paper is now ok from my side.

Minor modifications need. Authors can do that.